# Treatment Outcomes of Intravesical Botulinum Toxin A Injections on Patients with Interstitial Cystitis/Bladder Pain Syndrome

**DOI:** 10.3390/toxins14120871

**Published:** 2022-12-11

**Authors:** Wan-Ru Yu, Yuan-Hong Jiang, Jia-Fong Jhang, Wei-Chuan Chang, Hann-Chorng Kuo

**Affiliations:** 1Department of Nursing, Hualien Tzu Chi Hospital, Buddhist Tzu Chi Medical Foundation, Hualien 970, Taiwan; 2Institute of Medical Sciences, Tzu Chi University, Hualien 970, Taiwan; 3Department of Urology, Hualien Tzu Chi Hospital, Buddhist Tzu Chi Medical Foundation and Tzu Chi University, Hualien 970, Taiwan; 4Department of Medical Research, Buddhist Tzu Chi General Hospital, Hualien 970, Taiwan

**Keywords:** interstitial cystitis/bladder pain syndrome, botulinum toxin A injection, urine biomarkers, bladder inflammation

## Abstract

Botulinum toxin A (BoNT-A) is effective in reducing bladder hypersensitivity and increasing capacity through the effects of anti-inflammation in the bladder urothelium; however, studies on the treatment outcome of interstitial cystitis/bladder pain syndrome (IC/BPS) are lacking. We investigated the treatment outcome in IC/BPS patients receiving intravesical BoNT-A injections. This retrospective study included IC/BPS patients who had 100U BoNT-A intravesical injections in the past 20 years. The treatment outcomes at 6 months following the BoNT-A treatment were evaluated using the global response assessment (GRA) scale. The treatment outcomes according to the GRA scale include clinical symptoms, urodynamic parameters, cystoscopic characteristics, and urinary biomarkers, and it was these predictive factors for achieving satisfactory outcomes which were investigated. Among the 220 enrolled patients (180 women, 40 men) receiving BoNT-A injections, only 87 (40%) had significantly satisfactory treatment outcomes. The satisfactory group showed significantly larger voided volumes, and lower levels of both the urinary inflammatory protein MCP-1 and the oxidative stress biomarker 8-isoprostane in comparison to the unsatisfactory group. The IC severity and detrusor pressure are predictive factors of BoNT-A treatment outcomes. IC/BPS patients with less bladder inflammation showed satisfactory outcomes with intravesical BoNT-A injections. Patients with severe bladder inflammation might require more intravesical BoNT-A injections to achieve a satisfactory outcome.

## 1. Introduction

Interstitial cystitis/bladder pain syndrome (IC/BPS) is a urinary bladder disorder characterized by chronic pelvic pain, pressure, or discomfort, and is accompanied by urinary symptoms, including urinary frequency, nocturia, and urgency [1]. Its prevalence was reported to be 0.04% and 0.26% in Taiwan and Korea, respectively [1]. In other words, there are approximately 100,000 people with IC/BPS in Taiwan. The pathophysiology is still unclear, and patients with this condition have not achieved satisfactory treatment outcomes [2]. IC/BPS can be classified into Hunner’s (HIC) or non-Hunner’s (NHIC) ulcer types [1]. The most common pathological findings are urothelial denudation and bladder inflammation [2]. Failure to achieve full urothelial regeneration results in potential breaches in barrier function that may increase an individual’s susceptibility to infection or increase sensory fiber stimulation [3]. Nevertheless, multimodal therapies may be necessary to improve not only the physiological but also the psychological well-being of patients [4].

Previously, the American Urology Association (AUA) guidelines for IC/BPS recommended six steps of treatment; however, recently most guidelines do not recommend step-by-step treatments. Instead, multiple and simultaneous treatments were suggested [1,4,5,6]. Overall, the treatments include pain control, lifestyle modification, stress management, pelvic floor muscle therapy, oral therapies, intravesical therapies, and novel treatment for bladder inflammation [2].

As anti-inflammation and pain control is important for IC/BPS patients, the focus on bladder urothelium treatment is indispensable [7]. Botulinum toxin A (BoNT-A) is effective in reducing bladder hypersensitivity and increasing capacity through its anti-inflammatory and antinociceptive effects in the bladder urothelium [8]. BoNT-A not only reduces bladder pain effectively but it also increases bladder capacity in patients with cases of IC/BPS that are refractory to conventional therapy [9,10]. Furthermore, BoNT-A is capable of gradually decreasing bladder inflammation and enhancing urothelial repair, leading to symptomatic relief [11,12].

Due to IC/BPS’s refractory nature, further research and investigation is vitally important. In real-world practice, precision medicine can not only assist clinical doctors in identifying suitable treatment options but could also help IC/BPS patients achieve satisfactory treatment outcomes earlier. Recently, the correlations between urinary biomarkers and the pathophysiology of IC/BPS were explored [13]. However, the self-reported outcomes, according to the IC/BPS patients’ point of view, have not been investigated. Moreover, data relating to the effects of intravesical BoNT-A injections on improved self-reported treatment outcomes, and the predictive value of urinary biomarkers among IC/BPS patients are still lacking. Therefore, we aimed to investigate the treatment outcomes of intravesical BoNT-A injections in patients with IC/BPS in a real-life setting.

## 2. Results

In total, 220 patients with IC/BPS (180 women, and 40 men) who had received BoNT-A injections were enrolled. The mean age and IC/BPS duration were 54.1 ± 13.4 and 13.8 ± 10.1 years, respectively. The mean maximal bladder capacity (MBC) under anesthesia was 652 ± 204 mL; with 76 (35%), 114 (51.4%), and 20 (9.1%) patients having grade I glomerulation, grade II–III glomerulation, and HIC, respectively. The mean IC symptom index (ICSI) and mean IC problem index (ICPI) were 12.7 ± 3.7 and 12.0 ± 3.3 points, respectively. The mean numerical rating scale (NRS) score for bladder pain was 5.1 ± 2.7 points. In total, 124 (56%) patients had voiding dysfunction, which included bladder neck dysfunction (*n* = 8, 4%), dysfunctional voiding (*n* = 14, 6%), and poor external sphincter relaxation (102, 46%) whilst under videourodynamic study (VUDS), and all patients had storage dysfunction, which included bladder hypersensitivity (*n* = 198, 90%) and detrusor overactivity (*n* = 22, 10%); 214 (97%) patients had bladder pain or an intense urge to void during the potassium chloride (KCl) infusion test. In total, 180 (82%) patients reported having bladder, pelvic, or lower abdominal pain or discomfort (Table 1).

Among the 220 patients receiving BoNT-A injections, 133 (60%) reported unsatisfactory treatment outcomes which were evaluated using the global response assessment (GRA) at 6 months later, and 87 (40%) reported satisfactory treatment outcomes. The unsatisfactory group had a significantly higher rate of urinary tract symptoms (LUTS), including urinary frequency (88.9% vs. 69.9%), nocturia (87.5% vs. 74.2%), and urinary retention (6.9% vs. 1.1%) than the satisfactory group. However, the unsatisfactory group had lower urge incontinence (8.3% vs. 18.3%) rate. Nine (10%) patients in the satisfactory group underwent electrical coagulation of Hunner’s lesion. Moreover, as shown in Table 1, there were no significant differences in the subjective awareness items which included the ICSI and ICPI of the O’Leary–Sant symptom scale (OSS), and the NRS for bladder pain severity between the two groups, regardless of their self-reported treatment outcomes. However, the satisfactory group had a much larger voided volume (273 ± 130 vs. 236 ± 111 mL, *p* = 0.026) at the baseline. Moreover, there were no significant differences in age, sex, IC duration, pain severity, IC symptoms, problem severity, VUDS parameters, glomerulation grade, MBC, and KCl test results at the baseline between the two groups (Table 1).

In contrast, according to their bladder condition, we referred to a previous study that used statistical analysis of the receiver operating characteristic curve to predict satisfactory outcomes [11] and to further define both the MBCs greater or less than 760 mL and the phenotype divided on bladder capacity combined with the glomerulation grade. The unsatisfactory group was found to have a significantly smaller bladder capacity of <760 mL in comparison to the satisfactory group (*n* = 103, 77% vs. *n* = 53, 61%, *p* = 0.007). Interestingly, among 220 patients, excluding those with HIC, who had their urine collected for urinary biomarker analysis, the levels of the inflammatory biomarker monocyte chemoattractant protein-1 (MCP-1) and the oxidative stress biomarker 8-isoprostane were significantly higher in the unsatisfactory group (379 ± 517 vs. 229 ± 259, *p* = 0.031; 115 ± 245 vs. 44.8 ± 48.1, *p* = 0.017) (Table 2). Among the patients with satisfactory outcomes, 37% (*n* = 32) received intravesical BoNT-A injections more than four times, whereas, in the unsatisfactory group, only 27% (*n* = 36) received the BoNT-A injection more than four times.

Finally, to determine the IC/BPS patients’ subjective or objective influencing factors, including age, sex, IC duration, IC symptoms and problem severity, VUDS parameters, and phenotype, a supervised machine learning algorithm was used to predict the probability of an outcome. The results of the logistic regression indicated that IC/BPS symptoms and problem severity (odds ratio: 1.06, 95% CI: 1.01–1.13, *p* = 0.031) and detrusor pressure were significant predictors of treatment outcomes. Patients with grade 2–3 glomerulation and MBC < 760 mL might have poor treatment outcomes compared to those with grade 0 to 1 glomerulation and MBC ≥ 760 mL (odds ratio: 0.04, 95% CI: 0.17–0.97, *p* = 0.042) (Figure 1). However, adverse events including hematuria after injection in 6 (2.7%) patients, 4 (1.8%) urinary tract infections, and 36 (16%) mild dysuria cases were found, but these were without reports of urinary retention from any of the patients treated.

## 3. Discussion

In our study, only 40% of our IC/BPS patients showed symptomatic improvement. IC/BPS patients with less inflammatory bladder conditions would likely have satisfactory outcomes with intravesical BoNT-A treatment. The levels of inflammatory and oxidative stress urinary biomarkers, such as MCP-1 and 8-isoprostane, were higher in IC/BPS patients with unsatisfactory treatment outcomes. Moreover, voided volume and MBC also affect the IC/BPS patients’ treatment outcomes. IC/BPS symptom and problem severity and detrusor pressure are significant predictors of treatment outcomes in IC/BPS patients receiving intravesical BoNT-A injections. The IC/BPS patients with severe bladder inflammation might need more intravesical BoNT-A injections to attain a satisfactory outcome.

The urothelium is a stratified epithelium with three cell types (basal, intermediate, and superficial) and its functions include: forming a permeability barrier, acting as a sensory organ, and accommodating large volumes of urine [14]. The bladder urothelium is basically quiescent but regenerates readily upon injury [15]; the turnover rate of quiescent rodent urothelium is approximately once every 200 days [3]. Insufficient urothelial regeneration might decline the defense barrier and impact the barrier function whereby toxic substances or pathogens in the urine further stimulate local tissue inflammation [14], depolarize afferent nerve fibers, raise exposure to urinary toxins, incite chronic bladder inflammation, and aggravate sensory nerve activation, leading to chronic pain and insufficient or overabundant regeneration [3] which further results in higher levels of cytokine biomarkers in the urine.

The first documented therapeutic application of BoNT occurred in 1977. A purified BoNT (Oculinum©) was injected into the extra-ocular muscles to treat strabismus [16]. In 1988, Dykstra et al., first used BoNT in patients with lower urinary tract disorders to treat detrusor external sphincter dyssynergia [17]. Additionally, physicians also attempted to use intravesical BoNT-A injections as treatment for IC/BPS in 2003 [18]. The BoNT-A is one of the most powerful neurotoxins and inhibits the release of neurotransmitters from nerve fibers and urothelium [9]. The therapeutic effects might involve inhibiting the release of acetylcholine into the neuromuscular junctions of the detrusor muscle and anti-inflammatory responses [19]. Intravesical BoNT-A injections are listed in the AUA clinical guidelines as a fourth-line treatment option for IC/BPS [7].

In the present study, we reviewed the data of 220/521 IC/BPS patients receiving intravesical BoNT-A treatment over the past two decades, regardless of the number of BoNT-A injections each received. Only 40% of these patients showed symptomatic improvement, including those with long lasting disease durations and those who were refractory to medical treatments. Additionally, BoNT-A (100 U) is effective and safe as a treatment for IC/BPS [9]. Unfortunately, in the clinical practice, effective treatments for IC/BPS are lacking [1] and the pathophysiology of IC/BPS is also undetermined. However, previous guidelines believe the pathophysiology of NHIC might be multifactorial and include inflammation, post-infection autoimmune process, mast-cell activation, and urothelial dysfunction [1,7,9]. Furthermore, the bladder disorder could be first or secondary, which resulted from another cause [20]. However, bladder treatment combined with multimodal therapy is necessary [21].

Moreover, in the present study, we further explored the inflammatory and oxidative urinary biomarkers in these IC/BPS patients and found that patients without satisfactory treatment outcomes had higher MCP-1 and 8-isoprostane levels. In contrast, the patients reporting satisfactory treatment outcomes had significantly lower levels of inflammatory and oxidative stress urinary biomarkers. This leads us to presume that repeat BoNT-A injections not only reduce pain but also decrease bladder inflammation and improve IC symptoms and problems, resulting in better treatment outcomes. In the previous clinical trial using repeated BoNT-A injections, a higher success rate was found in patients receiving more than two injections over longer therapeutic durations [19]. In practice, when a patient reports no satisfactory outcome after receiving the BoNT-A injection, physicians might not recommend repeat injections and the patient probably would refuse such additional treatment without considering that the anti-inflammatory effect had not been attained during the initial treatment.

In summary, the etiology of IC/BPS might be affected by multiple factors, including a defective/damaged bladder urothelium, activation of C-fibers, neurogenic inflammation with mast cell activation, autoimmunity, occult infection, and pudendal nerve entrapment [1]. However, IC/BPS patients, with or without Hunner’s ulcers, had significantly higher levels of urine cytokine biomarkers, including interleukin-8 (IL-8), C-X-C motif chemokine ligand 10 (CXCL-10), brain-derived neurotrophic factor (BDNF), eotaxin, and regulated upon activation/normal T cell expressed and secreted (RANTES) than the general population [13]. This revealed that chronic inflammation might be the fundamental pathophysiology of IC/BPS. Therefore, bladder tissue apoptosis among IC/PBS patients might result from inflammatory signal upregulation [22]. This study, based on the point of view analysis of IC/BPS patients’ subjective treatment outcomes in correlation with objective factors, revealed that the unsatisfactory group has higher urine biomarker levels and lesser bladder capacities, indicating that they probably have more severe bladder inflammation and have not yet achieved optimal treatment effects. In these patients, repeat intravesical BoNT-A injections are required to achieve a satisfactory outcome. Regardless of the therapeutic options used by patients with IC/BPS in the past, in the future, precision medicine such as urine biomarkers, are expected to be used during bespoke personal treatment courses irrespective of physiological or psychological treatments, and longer, more complete treatment periods are also essential.

The major limitation of this study is its retrospective study design and single-center. Thus, further research is needed before more definite recommendations can be made, and in addition, longer follow-ups are also needed. Future studies should investigate the correlation between urinary biomarkers before and after BoNT-A treatment. Moreover, a consistent bladder volume must be ensured before collecting a urine sample and any invasive examinations must not be performed on patients for at least 48 h to ensure non-stimulation of the urothelium cell. Furthermore, allocating IC/BPS phenotype to suitable treatment options is another direction for our future efforts.

## 4. Conclusions

In this study, only 40% of IC/BPS patients had symptomatic improvement after intravesical BoNT-A injections. Patients with less inflammatory bladder conditions characterized by a larger bladder capacity, lower symptom severity, and lower urinary inflammatory and oxidative stress biomarker levels may predict satisfactory outcomes. Patients with severe bladder inflammation might require more intravesical BoNT-A injections to achieve a satisfactory outcome.

## 5. Materials and Methods

This study involved IC/BPS patients treated with 100 U BoNT-A from February 2000 to December 2021. All patients were diagnosed with IC/BPS in accordance with established IC/BPS characteristic symptoms and cystoscopic findings of glomerulations, petechiae, or mucosal fissures on anesthesia cystoscope hydrodistention [23]. Among the IC/BPS patients, lifestyle and behavioral modification, cystoscopic hydrodistention, intravesical hyaluronic acid instillation, or painkiller medications and treatment modalities were tried in at least two treatment modalities, but the IC symptoms persisted or relapsed. During the study period, some patients were undergoing BoNT-A injection clinical trials. However, we only recorded the outcomes of their first treatments.

All patients were screened thoroughly at the time of enrollment and were not enrolled if they failed the inclusion criteria of the European Society for the Study of Interstitial Cystitis [24]. This is a retrospective analysis of previous clinical trials of BoNT-A injections for patients with IC/BPS. In these clinical trials, the patient inclusion and exclusion criteria were the same (Appendix A).

The treatment outcomes at 6 months after the intravesical BoNT-A injection were evaluated using the GRA scale. Additionally, the IC symptoms were assessed using OSS, including ICSI and ICPI [25]. The ICSI and ICPI are two instruments used to determine the overall level of severity of each symptom and the significance of the problem from the patient’s perspective, respectively [26,27]. Both indices included four questions, one each for nocturia, frequency, urgency, and bladder-associated pain. The total ICSI scores range from 0 to 20. Each of the questions in the ICPI has five response options ranging from 0 to 4 with a maximum total ICPI score of 16, with higher scores indicating more severe IC/BPS symptoms and problem severity [26]. Patients were requested to rate their bladder symptoms as compared to that at baseline using a 7-point centered scale, from markedly (−3), moderately (−2), and slightly worse (−1), no change (0), to slightly (+1), moderately (+2), and markedly improved (+3). Patients with moderately and markedly improved results after treatment were considered to have satisfactory treatment outcomes. Otherwise, the treatment outcome was considered unsatisfactory [19].

VUDS was performed before the BoNT-A injection using the multichannel urodynamic system (Life-Tech, Stafford, TX, USA) and a C-arm fluoroscope (Toshiba, Tokyo, Japan). According to International Continence Society recommendations, this study’s descriptions and terminologies all follow the compliance criteria [28]. Based on the characteristic VUDS findings, such as the first sensation of filling, first desire to void full bladder sensation, cystometric bladder capacity, detrusor pressure at maximum flow rate, maximum flow rate, voided-volume, and post-void residual (PVR), patients would be diagnosed with hypersensitive bladder, detrusor overactivity, voiding dysfunction, poor pelvic floor muscle relaxation, or intrinsic sphincter deficiency [29]. The KCl test was considered positive if there was bladder pain or an intense urge to void during the KCl infusion after the emptying of the residual urine [30]. was Patients with increased bladder sensation and positive KCl sensitivity tests were encouraged to undergo Cystoscopic hydrodistention. The VUDS was performed to verify the diagnosis of IC/BPS at baseline and recognize other bladder conditions that resemble IC/BPS. A duplicate VUDS was performed 6 months after the primary BoNT-A injection to estimate the bladder condition after treatment and as an action for instigating subsequent treatment.

After cystoscopic hydrodistention, the patients were treated with consecutive bladder-targeting medications for bladder pain, including nonsteroidal-inflammatory drugs, cyclooxygenase-2 inhibitors, antimuscarinics, alpha-blockers, intravesical hyaluronic acid installations, and 4th line of intravesical BoNT-A injections, according to AUA guideline recommendations [20].

BoNT-A medicinal liquid constituted a vial of onabotulinumtoxin A (100 U) diluted with 10 mL 0.9% saline. Twenty injections were performed with this BoNT-A liquid, lead 5-U BoNT-A in each injection site. For the bladder’s posterior and lateral walls, an injection needle was inserted approximately 1 mm in the urothelium, sparing the trigone, using a 23-gauge needle and rigid cystoscopic injection instrument (22 Fr, Richard Wolf, and Knittlingen, Germany). After the BoNT-A injections, cystoscopic hydrodistention was performed under slowly dripping 0.9% saline to an intravesical pressure of 80 cm fluid for 15 min. The MBC and glomerulation grade under hydrodistention was also recorded after intravesical pressure release [9]. Based on the appearance of glomerulations for none, less than half, more than half, and more than half and during serious waterfall bleeding of the bladder wall, if patients have Hunner’s lesion combined with or without glomerulations were classified as ulcer-type IC/BPS. After that, the glomerulation grade was classified into 0, 1, 2, and 3 [11]. After the BoNT-A treatment, a 14-Fr indwelling urinary catheter was inserted overnight and removed the next day. An antibiotic (cephradine 500 mg every 6 h) was routinely prescribed for a week, and patients visited the outpatient clinic 2 weeks after treatment, followed by monthly visits to the outpatient clinic for outcome assessment. The primary endpoint was 6 months after the BoNT-A injection.

We not only analyzed the patients’ subjective and objective characteristics and VUDS parameters but also collected urine specimens to further analyze the urinary biomarkers at baseline. The urinary biomarkers collected included interleukin-8 (IL-8), CXCL10, MCP-1, BDNF, eotaxin, Interleukin 6 (IL-6), RANTES (also known as CCL5), prostaglandin E2, tumor necrosis factor-alpha, 8-hydroxy-2-deoxyguanosine, 8-isoprostane, and total antioxidant capacity [13]. In brief, before cystoscopic hydrodistention, all patients would collect 50 mL urine samples, obtained by self-urination, when patients had a full bladder sensation, and also excluded those with confirmed urinary tract infections. Before transferring to the laboratory, the urine samples were placed on ice. However, HICs would be excluded from further analysis considering the different pathology [13,31].

### Statistical Analysis

Statistical analysis was performed using SPSS version 25 (IBM, Armonk, NY, USA); a *p* value of <0.05 was considered statistically significant. Data were expressed as mean and standard deviations for continuous variables, and categorical variables were presented as counts and proportions. Between-group statistical comparisons were tested for categorical variables using either Pearson’s chi-square or Fisher’s exact test, and additionally using an independent t-test for continuous variables, excluding outliers of urinary biomarkers.

To examine the association of treatment outcomes, logistic regression models were further estimated for subjective and objective factors in an attempt to predict the factors which influence the expected treatment outcomes. Logistic regression was used to analyze the relationship between IC/BPS patients’ characteristics, VUDS parameters, and IC/BPS bladder phenotype and self-reported satisfactory treatment outcomes.

## Figures and Tables

**Figure 1 toxins-14-00871-f001:**
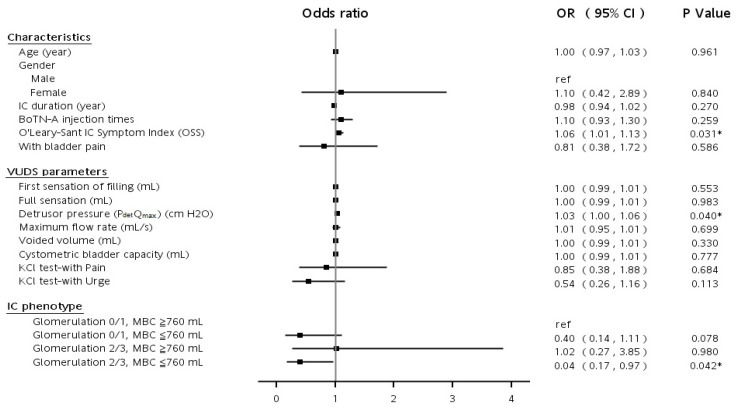
The forest plot results from the logistic regression for predicting satisfactory treatment outcome (GRA ≥ 2) effect factors. * significant *p* < 0.05.

**Table 1 toxins-14-00871-t001:** IC/BPS patients’ characteristics and VUDS parameters according to the treatment outcome (*n =* 220).

Variable			Total (*n* = 220)	Unsatisfactory Outcome GRA ≤ 1 (*n =* 133)	Satisfactory Outcome GRA ≥ 2 (*n =* 87)	*p* Value
**Characteristics**						
Age (years)			54.1 ± 13.4	53.9 ± 14.2	55.2 ± 12.3	0.499
Gender (%)	Men		40 (18%)	25 (19%)	15 (17%)	0.458
	Women		180 (82%)	108 (81%)	72 (83%)	
IC duration (years)			13.8 ± 10.1	14.5 ± 10.9	12.6 ± 8.6	0.176
Numerical rating pain scale			5.1 ± 2.7	4.9 ± 2.7	5.3 ± 2.6	0.332
IC symptoms index (ICSI)			12.7 ± 3.7	12.5 ± 3.7	13.1 ± 3.8	0.307
IC problem index (ICPI)			12.0 ± 3.3	11.6 ± 3.2	12.5 ± 3.4	0.052
O’Leary-Sant IC Symptom Index (OSS)			24.7 ± 6.6	24.1 ± 6.5	25.6 ± 6.8	0.118
**VUDS parameters**						
First sensation of filling (mL)			117 ± 52.4	116 ± 51.9	118 ± 53.4	0.751
Full sensation (mL)			182 ± 73.5	180 ± 74.7	184 ± 71.9	0.719
Cystometric bladder capacity (mL)			276 ± 114	267 ± 105	289 ± 126	0.167
Detrusor pressure (P_det_Q_max_) (cm H_2_O)			20.8 ± 12.8	20.8 ± 12.3	21.0 ± 13.3	0.572
Maximum flow rate (mL/s)			12.3 ± 5.7	11.8 ± 5.5	13.1 ± 6.0	0.572
Voided-volume (mL)			251 ± 121	236 ± 111	273 ± 130	0.026 *
Post-void residual (mL)			25.9 ± 51.6	28.8 ± 53.6	21.5 ± 48.4	0.308
KCl test—Pain (%)			167 (77.3%)	101 (76%)	66 (76%)	0.531
KCl test—Urge (%)			68 (31.5%)	47 (35%)	21 (24%)	0.056
Maximum bladder capacity (mL)			652 ± 204	635 ± 199	680 ± 209	0.113
Glomerulation grade		Grade 1	86 (39.5%)	48 (36%)	38 (44%)	0.173
		Grade 2	83 (37.3%)	55 (41%)	28 (32%)	
		Grade 3	31 (14.1%)	20 (15%)	11 (13%)	
Hunner’s lesion IC			20 (9.1%)	11 (8%)	9 (10%)	

* Significant *p* < 0.05.

**Table 2 toxins-14-00871-t002:** Bladder condition and urinary biomarkers of the IC/BPS patients stratified according to the treatment outcome (*n =* 220).

Variable	Total (*n* = 220)	Unsatisfactory Outcome GRA ≤ 1 (*n =* 133)	Satisfactory Outcome GRA ≥ 2 (*n =* 87)	*p* Value
**Bladder condition (%)**				
Maximal bladder capacity < 760 (mL)	156 (70.9%)	103 (77%)	53 (61%)	0.007 *
Glomerulation grade 2/3	114 (51.8%)	75 (56%)	39 (45%)	0.062
Hunner’s lesion IC	20 (9.1%)	11 (8%)	9 (10%)	0.097
Glomerulation 0/1, MBC ≥ 760 mL	45 (20.5%)	20 (15%)	25 (29%)	
Glomerulation 0/1, MBC < 760 mL	38 (17.3%)	26 (20%)	12 (14%)	
Glomerulation 2/3, MBC ≥ 760 mL	18 (8.2%)	10 (8%)	8 (10%)	
Glomerulation 2/3, MBC < 760 mL	99 (45%)	66 (47%)	33 (38%)	
**Urine biomarker (exclude HIC)**				
IL-8	23.7 ± 71.2	18.2 ± 24.2	31.6 ± 108	0.291
CXCL 10	16.4 ± 13.2	19.7 ± 46.7	11.5 ± 37.2	0.289
MCP-1	318 ± 436	379 ± 517	229 ± 259	0.029 *
BDNF	1.48 ± 9.35	2.04 ± 12.1	0.64 ± 0.29	0.398
Eotaxin	7.05 ± 7.63	7.83 ± 8.04	5.92 ± 6.89	0.157
IL-6	8.54 ± 53.7	11.2 ± 67.5	4.71 ± 19.0	0.496
MIP-1β	1.70 ± 8.88	1.79 ± 2.61	1.56 ± 5.23	0.728
RANTES	10.6 ± 61.1	15.5 ± 79	3.59 ± 5.06	0.273
TNF-α	6.91 ± 35.6	8.77 ± 44.6	4.19 ± 14.7	0.468
PGE2	329 ± 346	370 ± 382	269 ± 278	0.100
8-OHdG	34.3 ± 26.5	36.9 ± 29.6	30.5 ± 20.9	0.168
8-Isoprostane	86.5 ± 194	115 ± 245	44.8 ± 48.1	0.015 *
TAC	1214 ± 1203	1323 ± 1288	1058 ± 1063	0.218

MBC: Maximal bladder capacity, IL-8: interleukin-8, CXCL10: C-X-C motif chemokine ligand 10, MCP-1: monocyte chemoattractant protein-1, BDNF: brain-derived neurotrophic factor, IL-6: Interleukin 6, RANTES: regulated upon activation/normal T cell expressed and secreted, PGE2: prostaglandin E2, TNF-α: tumor necrosis factor-alpha, 8-OHdG: 8-hydroxy-2-deoxyguanosine, TAC: total antioxidant capacity; * significant *p* < 0.05.

## Data Availability

Data are available if requested from the corresponding authors.

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
