# Peer review of "Treatment Outcomes of Intravesical Botulinum Toxin A Injections on Patients with Interstitial Cystitis/Bladder Pain Syndrome"

_toxins, 2022, doi:10.3390/toxins14120871_

Round 1

Reviewer 1 Report

Thank you for the opportunity to review this novel manuscript entitled “Treatment outcome of intravesical botulinum toxin A injection on patients with interstitial cystitis/bladder pain syndrome: A real-life satisfaction and predictive factors”. I recognize that this paper is very interesting due to the predictive approach to the outcome of treatment of interstitial cystitis/ bladder pain syndrome patients with intravesical botulinum toxin type A which may be a key focus during this therapeutic approach. This type of prediction model could be used in order to avoid injuries or overtreatments in several vital body parts. Thus, this study may be a starting point in order to develop future research studies about new prediction models. 

-Abstract: The abstract is concise and adequate, according to the journal rules.

-Introduction: The background is very deep including the needed aspects to justify this study. 

-Results; This section summarize in a correct way the main findings of the study and the figures help the readers to follow the paper in a graphic manner.

-Discussion: Good discussion of the main findings highlighting clinical practice applications.

-Conclusion: This section is conclusive and summarizes in a correct way the treatment outcomes at six months after the intravesical botulinum toxin injection and the main study findings.

-Material and Methods: The methodology, sample size and statistical analysis of this retrospective study are based on prior studies in a correct way.

Author Response

Dear Reviewers: Thank you for the constructive comments. We have revised the manuscript according to your suggestions. The followings are the point-to-point replies to the individual comment

Reviewer 1

Comments and Suggestions for Authors

Thank you for the opportunity to review this novel manuscript entitled “Treatment outcome of intravesical botulinum toxin A injection on patients with interstitial cystitis/bladder pain syndrome: A real-life satisfaction and predictive factors”. I recognize that this paper is very interesting due to the predictive approach to the outcome of treatment of interstitial cystitis/ bladder pain syndrome patients with intravesical botulinum toxin type A which may be a key focus during this therapeutic approach. This type of prediction model could be used in order to avoid injuries or over treatments in several vital body parts. Thus, this study may be a starting point in order to develop future research studies about new prediction models.

-Abstract: The abstract is concise and adequate, according to the journal rules.

-Introduction: The background is very deep including the needed aspects to justify this study.

-Results; This section summarize in a correct way the main findings of the study and the figures help the readers to follow the paper in a graphic manner.

-Discussion: Good discussion of the main findings highlighting clinical practice applications.

-Conclusion: This section is conclusive and summarizes in a correct way the treatment outcomes at six months after the intravesical botulinum toxin injection and the main study findings.

-Material and Methods: The methodology, sample size and statistical analysis of this retrospective study are based on prior studies in a correct way.

Reply: Thank you for the comments. We appreciate that you understand how important this disease is. Interstitial cystitis/ bladder pain syndrome (IC/BPS) is an unknown disease until now and impacts huge negative influences. Clinically, IC/BPS also correlates with autonomic nervous system diseases such as fibromyalgia, irritable bowel, etc…We will try to explore IC/BPS and look forward to letting more doctors realize to help patients with IC/BPS, someone the world leaves behind.

Reviewer 2 Report

The manuscript entitled "Treatment outcome of intravesical botulinum toxin A injection on patients with interstitial cystitis/bladder pain syndrome – a real-life satisfaction and predictive factors" is well written and worthy of in Toxins, after some minor improvements.

I suggest the authors consider modifying the manuscript's title, deleting "– A real-life satisfaction and predictive factors".

Abstract

Page 1, line 7. Please delete the term real world.

Introduction

Page 1, lines 34-35. The authors reported the prevalence of IC/BPS in Taiwan and Korea. Are available further epidemiological data to better describe the burden of the disease? Please include them in the text.

Table 1

In table 1, the authors include "comorbidities". Please modify the variable as "number of comorbidities". In addition, the text contains no information on comorbidities and their impact on the patients enrolled in the study. If this information is relevant to the study should be included in the text; otherwise, the authors should consider deleting the variable in table 1. 

Discussion

Page 5, line 159 and page 6, line 188. Please change doctors with physicians.

Page 6, line 171. Please modify the term real-word with clinical practice. 

Conclusion

Although I suggested that the authors modify the title of the manuscript, I think that the authors should include some sentences on the predictive factors in the text.

Material and methods

Page 7, lines 234-236. The inclusion criteria adopted in the study are those established by the European Society for the Study of Interstitial Cystitis. I suggest the authors evaluate the opportunity to summarize the inclusion criteria in a new table to facilitate the manuscript's readability. 

Page 7, lines 236-237. The GRA scale should be defined in the text or a table. 

Author Response

Dear Reviewers: Thank you for the constructive comments. We have revised the manuscript according to your suggestions. The followings are the point-to-point replies to the individual comment

Reviewer 2

Comments and Suggestions for Authors

The manuscript entitled "Treatment outcome of intravesical botulinum toxin A injection on patients with interstitial cystitis/bladder pain syndrome – a real-life satisfaction and predictive factors" is well written and worthy of in Toxins, after some minor improvements.

I suggest the authors consider modifying the manuscript's title, deleting "– A real-life satisfaction and predictive factors".

Reply: Thank you for the comments. The title has been deleting the words "– A real-life satisfaction and predictive factors". (Line 3)

Abstract

Page 1, line 7. Please delete the term real world.

Reply: Thank you for the comments. The text has been deleted the term real world. (Line 6)

Introduction

Page 1, lines 34-35. The authors reported the prevalence of IC/BPS in Taiwan and Korea. Are available further epidemiological data to better describe the burden of the disease? Please include them in the text.

Reply: Thank you for the comments. Homma reported The Taiwan National Database showed the prevalence was 40.2 out of 100,000 in 2013, it’s about 100 thousand people in Taiwan until now, and this manuscript refer to the article and adjust the same unit in this text. We have added the prevalence data in the text. (Line 34)

Table 1

In table 1, the authors include "comorbidities". Please modify the variable as "number of comorbidities". In addition, the text contains no information on comorbidities and their impact on the patients enrolled in the study. If this information is relevant to the study should be included in the text; otherwise, the authors should consider deleting the variable in table 1.

Reply: Thank you for the comments. The comorbidities data in Table 1 had been deleted.

Discussion

Page 5, line 159 and page 6, line 188. Please change doctors with physicians.

Reply: Thank you for the comments. The term has been revised, accordingly (Line 160, 189).

Page 6, line 171. Please modify the term real-word with clinical practice.

Reply: Thank you for the comments. The term has been revised, accordingly (Line 172).

Conclusion

Although I suggested that the authors modify the title of the manuscript, I think that the authors should include some sentences on the predictive factors in the text.

Reply: Thank you for the comments. We have revised the conclusion and add predictive factors of successful outcome, accordingly. (Line 220-225)

Material and methods

Page 7, lines 234-236. The inclusion criteria adopted in the study are those established by the European Society for the Study of Interstitial Cystitis. I suggest the authors evaluate the opportunity to summarize the inclusion criteria in a new table to facilitate the manuscript's readability.

Reply: Thank you for the comments. This is a retrospective analysis of previous clinical trials of BoNT-A injection for patients with IC/BPS. In these clinical trials, the patient inclusion and exclusion criteria were the same. We have added the inclusion and exclusion criteria of our study in the appendix of this manuscript. (Line 387- )

Page 7, lines 236-237. The GRA scale should be defined in the text or a table.

Reply: Thank you for the comments. The GRA scale had been defined in the Method section. (Lines 251-257).

Round 2

Reviewer 2 Report

The manuscript can be published. At page 6 line 172 I suggest to delete the words "real-word with" and add the term "the". I think that this modification can be done at level of proof reading.